# Preference Completion from Partial Rankings

**Suriya Gunasekar**
University of Texas, Austin, TX, USA
suriya@utexas.edu

**Oluwasanmi Koyejo**
University of Illinois, Urbana-Champaign, IL, USA
sanmi@illinois.edu

**Joydeep Ghosh**
University of Texas,Austin, TX, USA
ghosh@ece.utexas.edu

## Abstract

We propose a novel and efficient algorithm for the collaborative preference completion problem, which involves jointly estimating individualized rankings for a set of entities over a shared set of items, based on a limited number of observed affinity values. Our approach exploits the observation that while preferences are often recorded as numerical scores, the predictive quantity of interest is the underlying rankings. Thus, attempts to closely match the recorded scores may lead to overfitting and impair generalization performance. Instead, we propose an estimator that directly fits the underlying preference order, combined with nuclear norm constraints to encourage low–rank parameters. Besides (approximate) correctness of the ranking order, the proposed estimator makes no generative assumption on the numerical scores of the observations. One consequence is that the proposed estimator can fit any consistent partial ranking over a subset of the items represented as a directed acyclic graph (DAG), generalizing standard techniques that can only fit preference scores. Despite this generality, for supervision representing total or blockwise total orders, the computational complexity of our algorithm is within a $\log$ factor of the standard algorithms for nuclear norm regularization based estimates for matrix completion. We further show promising empirical results for a novel and challenging application of collaboratively ranking of the associations between brain–regions and cognitive neuroscience terms.

## 1  Introduction

Collaborative preference completion is the task of jointly learning bipartite (or dyadic) preferences of set of entities for a shared list of items, e.g., user–item interactions in a recommender system [14; 22]. It is commonly assumed that such entity–item preferences are generated from a small number of latent or hidden factors, or equivalently, the underlying preference value matrix is assumed to be low rank. Further, if the observed affinity scores from various explicit and implicit feedback are treated as exact (or mildly perturbed) entries of the unobserved preference value matrix, then the preference completion task naturally fits in the framework of low rank matrix completion [22; 38]. More generally, low rank matrix completion involves predicting the missing entries of a low rank matrix from a vanishing fraction of its entries observed through a noisy channel. Several low rank matrix completion estimators and algorithms have been developed in the literature, many with strong theoretical guarantees and empirical performance [6; 32; 21; 28; 38; 10].

Recent research in the preference completion literature have noted that using a matrix completion estimator for collaborative preference estimation may be misguided [11; 33; 23] as the observed entity–item affinity scores from implicit/explicit feedback are potentially subject to systematic monotonic transformations arising from limitations in feedback collection, e.g., quantization and

inherent biases. While simple user biases and linear transofmations can be handled within a low rank matrix framework, more complex transformations like quantization can potentially increase the rank of the observed preference score matrix significantly, thus adversely affecting recovery using standard low rank matrix completion [13]. Further, despite the common practice of measuring preferences using numerical scores, predictions are most often deployed or evaluated based on the item ranking e.g. in recommender systems, user recommendations are often presented as a ranked list of items without the underlying scores. Indeed several authors have shown that favorable empirical/theoretical performance in mean square error for the preference matrix often does not translate to better performance when performance is measured using ranking metrics [11; 33; 23]. Thus, collaborative preference estimation may be better posed as a collection of coupled *learning to rank (LETOR)* problems [25], where we seek to jointly learn the preference rankings of a set of entities, by exploiting the low dimensional latent structure of the underlying preference values.

This paper considers preference completion in a general collaborative LETOR setting. Importantly, while the observations are assumed to be reliable indicators for relative preference ranking, their numerical scores may be quite deviant from the ground truth low rank preference matrix. Therefore, we aim at addressing preference completion under the following generalizations:

1. In a simple setting, for each entity, a score vector representing the its observed affinity interactions is assumed to be generated from an *arbitrary monotonic transformation* of the corresponding entries of the ground truth preference matrix. We make *no* further generative assumptions on observed scores beyond monotonicity with respect to the underlying low rank preference matrix.
2. We also consider a more general setting, where observed preferences of each entity represent specifications of a *partial ranking* in the form of a directed acyclic graph (DAG) – the nodes represent a subset of items, and each edge represents a strict ordering between a pair of nodes. Such rankings may be encountered when the preference scores are consolidated from multiple sources of feedback, e.g., comparative feedback (pairwise or listwise) solicited for independent subsets of items. This generalized setting cannot be handled by standard matrix completion without some way of transforming the DAG orderings into a score vector.

Our work is in part motivated by an application to neuroimaging meta-analysis as outlined in the following. Cognitive neuroscience aims to quantify the link between brain function with behavior. This interaction is most often measured in humans using Functional Magnetic Resonance Imaging (fMRI) experiments that measure brain activity in response to behavioral tasks. After analysis, the conclusions are often summarized in neuroscience publications which include a table of brain locations that are most actively activated in response to an experimental stimulus. These results can then be synthesized using meta-analysis techniques to derive accurate predictions of brain activity associated with cognitive terms (also known as forward inference) and prediction of cognitive terms associated with brain regions (also known as reverse inference). For our study, we used data from neurosynth [36] - a public repository[1] which automatically scrapes information on published associations between brain regions and terms in cognitive neuroscience experiments.

The key contributions of the paper are summarized below.

- We propose a convex estimator for low rank preference completion using limited supervision, addressing: (a) arbitrary monotonic transformations of preference scores; and (b) partial rankings over items (Section 3.1). We derive generalization error bounds for a surrogate ranking loss that quantifies the trade–off between data–fit and regularization (Section 5).
- We propose efficient algorithms for the estimate under total and partially ordered observations. In the case of total orders, in spite of increased generality, the computational complexity of our algorithm is within a $\log$ factor of the standard convex algorithms for matrix completion (Section 4).
- The proposed algorithm is evaluated for a novel application of identifying associations between brain–regions and cognitive terms from the neurosynth dataset [37] (Section 6). Such a large scale meta-analysis synthesizing information from the literature and related tasks has the potential to lead to novel insights into the role of brain regions in cognition and behavior.

## 1.1 Notation

For a matrix $M \in \mathbb{R}^{d_1 \times d_2}$, let $\sigma_1 \geq \sigma_2 \geq \ldots$ be singular values of $M$. Then, nuclear norm $\|M\|_* = \sum_i \sigma_i$, operator norm $\|M\|_{\mathrm{op}} = \sigma_1$, and Frobenius norm $\|M\|_F = \sqrt{\sum_i \sigma_i^2}$. Let

$[N] = \{1, 2, \ldots, N\}$. A vector or a set $x$ indexed by $j \in [N]$ is sometimes denoted as $(x_j)_{j=1}^N$ or simply $(x_j)$ whenever $N$ is unambiguous. Let $\Omega \subset [d_1] \times [d_2]$ denote a subset of indices of a matrix in $\mathbb{R}^{d_1 \times d_2}$. For $j \in [d_2]$, let $\Omega_j = \{(i', j') \in \Omega : j' = j\} \subset \Omega$ denotes the subset of entries in $\Omega$ from the $j^{\text{th}}$ column. Given $\Omega = \{(i_s, j_s) : s = 1, 2, \ldots, |\Omega|\}$, $\mathcal{P}_\Omega : X \to (X_{i_s j_s})_{s=1}^{|\Omega|} \in \mathbb{R}^{|\Omega|}$ is the linear subsampling operator, and $\mathcal{P}_\Omega^* : \mathbb{R}^{|\Omega|} \to \mathbb{R}^{d_1 \times d_2}$ is its adjoint, i.e $\langle y, \mathcal{P}_\Omega(X) \rangle = \langle X, \mathcal{P}_\Omega^*(y) \rangle$. For conciseness, we sometimes use the notation $X_\Omega$ to denote $\mathcal{P}_\Omega(X)$.

## 2    Related Work

**Matrix Completion:**   Low rank matrix completion has an extensive literature; a few examples include [22; 6; 21; 28] among several others. However, the bulk of these works including those in the context of ranking/recommendation applications focus on (a) fitting the observed numerical scores using squared loss, and (b) evaluating the results on parameter/rating recovery metrics such as root mean squared error (RMSE). The shortcomings of such estimators and results using squared loss in ranking applications have been studied in some recent research [12; 11]. Motivated by collaborative ranking applications, there has been growing interest in addressing matrix completion within an explicit LETOR framework. Weimer et al. [35] and Koyejo et al. [23] propose estimators that involve non–convex optimization problems and their algorithmic convergence and generalization behavior are not well understood. Some recent works provide parameter recovery guarantees for pairwise/listwise ranking observations under specific probabilistic distributional assumptions on the observed rankings [31; 26; 29]. In comparison, the estimators and algorithms in this paper are agnostic to the generative distribution, and hence have much wider applicability.

**Learning to rank (LETOR):** LETOR is a structured prediction task of rank ordering relevance of a list of items as a function of pre–selected features [25]. Currently, leading algorithms for LETOR are listwise methods [9] (as is the approach taken in this paper), which fully exploit the ranking structure of ordered observations, and offer better modeling flexibility compared to the pointwise [24] and pairwise methods [16; 18]. A recent listwise LETOR algorithm proposed the idea of monotone retargeting (MR) [2], which elegantly addresses listwise learning to rank (LETOR) task while maintaining the relative simplicity and scalability of pointwise estimation. MR was further extended to incorporate margins in the *margin equipped monotonic retargeting (MEMR)* formulation [1] to preclude trivial solutions that arise from scale invariance of the initial MR estimate in Acharyya et al. [2]. The estimator proposed in the paper is inspired from the the idea of MR and will be revisited later in the paper. In collaborative preference completion, rather than learning a functional mapping from features to ranking, we seek to exploit the low rank structure in jointly modeling the preferences of a collection of entities without access to preference indicative features.

**Single Index Models (SIMs)** Finally, literature on monotonic single index models (SIMs) also considers estimation under unknown monotonic transformations [17; 20]. However, algorithms for SIMs are designed to solve a harder problem of exactly estimating the non–parametric monotonic transformation and are evaluated for parameter recovery rather than the ranking performance. In general, with no further assumptions, sample complexity of SIM estimators lends them unsuitable for high dimensional estimation. The existing high dimensional estimators for learning SIMs typically assume Lipschitz continuity of the monotonic transformation which explicitly uses the observed score values in bounding the Lipsciptz constant of the monotonic transformation [19; 13]. In comparison, our proposed model is completely agnostic to the numerical values of the preference scores.

## 3    Preference Completion from Partial Rankings

Let the unobserved true preference scores of $d_2$ entities for $d_1$ items be denoted by a rank $r \ll \min\{d_1, d_2\}$ matrix $\Theta^* \in \mathbb{R}^{d_1 \times d_2}$. For each entity $j \in [d_2]$, we observe a partial or total ordering of preferences for a subset of items denoted by $\mathcal{I}_j \subset [d_1]$. Let $n_j = |\mathcal{I}_j|$ denotes the number of items over which relative preferences of entity $j$ are observed, so that $\Omega_j = \{(i, j) : i \in \mathcal{I}_j\}$ denotes the entity-item index set for $j$, and $\Omega = \bigcup_j \Omega_j$ denotes the index set collected across entities. Let $\mathbb{P}_\Omega$ denote the sampling distribution for $\Omega$. The observed preferences of entity $j$ are typically represented by a listwise preference score vector $y^{(j)} \in \mathbb{R}^{n_j}$.

$$\forall j \in [d_2], \, y^{(j)} = g_j(\mathcal{P}_{\Omega_j}(\Theta^* + W)), \tag{1}$$

where each $(g_j)$ are an *arbitrary and unknown monotonic transformations*, and $W \in \mathbb{R}^{d_1 \times d_2}$ is some non–adversarial noise matrix sampled from the distribution $\mathbb{P}_W$. The *preference completion task* is to estimate a unseen rankings within each column of $\Theta^*$ from a subset of orderings $(\Omega_j, y^{(j)})_{j \in [d_2]}$.

As $(g_j)$ are arbitrary, the exact values of $(y^{(j)})$ are inconsequential, and the observed preference order can be specified by a constraint set parameterized by a margin parameter $\epsilon$ as follows:

**Definition 1 ($\epsilon$–margin Isotonic Set)** The following set of vectors are isotonic to $y \in \mathbb{R}^n$ with an $\epsilon > 0$ margin parameter:
$$\mathcal{R}_{\downarrow\epsilon}^n(y) = \{x \in \mathbb{R}^n : \forall \, i, k \in [n], \, y_i < y_k \Rightarrow x_i \leq x_k - \epsilon\}.$$

In addition to score vectors, isotonic sets of the form $\mathcal{R}_{\downarrow\epsilon}^n(y)$ are equivalently defined for any DAG $y = \mathcal{G}([n], E)$ which denotes a partial ranking among the vertices, with the convention that $(i,k) \in E \Rightarrow \forall x \in \mathcal{R}_{\downarrow\epsilon}^n(y), \, x_i \leq x_k - \epsilon$. We note from Definition 1 that ties are *not* broken at random, e.g., if $y_{i_1} = y_{i_2} < y_k$, then $\forall x \in \mathcal{R}_{\downarrow\epsilon}^n(y), x_{i_1} \leq x_k - \epsilon, x_{i_2} \leq x_k - \epsilon$, but no particular ordering between $x_{i_1}$ and $x_{i_2}$ is specified.

Let $y_{(k)}$ denote the $k^{\text{th}}$ smallest entry of $y \in \mathbb{R}^n$. We distinguish between three special cases of an observation $y$ representing a partial ranking over $[n]$.
(A) *Strict Total Order*: $y_{(1)} < y_{(2)} < \ldots < y_{(n)}$.
(B) *Blockwise Total Order*: $y_{(1)} \leq y_{(2)} \leq \ldots \leq y_{(n)}$, with $K \leq n$ unique values.
(C) *Arbitrary DAG*: Partial order induced by a DAG $y = \mathcal{G}([n], E)$.

## 3.1 Monotone Retargeted Low Rank Estimator

Consider any scalable pointwise learning algorithm that fits a model to exact preferences scores. Since no generative model (besides monotonicity) is assumed for the raw numerical scores in the observations, in principle, the scores $y^{(j)}$ for entity $j$ can be replaced or *retargeted* to any ranking-preserving scores, i.e., by any vector in $\mathcal{R}_{\downarrow\epsilon}^{n_j}(y^{(j)})$. Monotone Retargeting (MR) [2] exploits this observation to address the combinatorial listwise ranking problem [25] while maintaining the relative simplicity and scalability of pointwise estimates (regression). The key idea in MR is to alternately fit a pointwise algorithm to current relevance scores, and *retarget* the scores by searching over the space of all monotonic transformations of the scores. Our approach extends and generalizes monotone retargeting for the preference prediction task.

We begin by motivating an algorithm for the noise free setting, where it is clear that $\Theta^*_{\Omega_j} \in \mathcal{R}_{\downarrow\epsilon}^{n_j}(y^{(j)})$, so we seek to estimate a candidate preference matrix $X$ that is in the intersection of (a) the data constraints from the observed preference rankings $\{X_{\Omega_j} \in \mathcal{R}_{\downarrow\epsilon}^{n_j}(y^{(j)})\}$, and (b) the model constraints – in this case low rankness induced by constraining the nuclear norm $\|X\|_*$. For robust estimation in the presence of noise, we may extend the noise free approach by incorporating a soft penalty on constraint violations. Let $z \in \mathbb{R}^{|\Omega|}$, and with slight abuse of notation, let $z_{\Omega_j} \in \mathbb{R}^{n_j}$ denote vector of the entries of $z \in \mathbb{R}^{|\Omega|}$ corresponding to $\Omega_j \subset \Omega$. Upon incorporating the soft penalties, the monotone retargeted low rank estimator is given by:

$$\widehat{\mathcal{X}} = \underset{X}{\mathrm{Argmin}} \min_{z \in \mathbb{R}^{|\Omega|}} \lambda \|X\|_* + \frac{1}{2} \|z - \mathcal{P}_\Omega(X)\|_2^2 \quad \text{s.t.} \forall j, \; z_{\Omega_j} \in \mathcal{R}_{\downarrow\epsilon}^{n_j}(y^{(j)}), \tag{2}$$

where the parameter $\lambda$ controls the trade–off between nuclear norm regularization and data fit, and $\widehat{\mathcal{X}}$ is the set of minimizers of (2). We note that $\mathcal{R}_{\downarrow\epsilon}^n(y)$ is convex, and $\forall \lambda \geq 1$, the scaling $\lambda \mathcal{R}_{\downarrow\epsilon}^n(y) = \{\lambda x \; \forall \; x \in \mathcal{R}_{\downarrow\epsilon}^n(y)\} \subseteq \mathcal{R}_{\downarrow\epsilon}^n(y)$. The above estimate can be computed using efficient convex optimization algorithms and can handle arbitrary monotonic transformation of the preference scores, thus providing higher flexibility compared to the standard matrix completion.

Although (2) is specified in terms of two parameters, due to the geometry of the problem, it turns out that $\lambda$ and $\epsilon$ are not jointly identifiable, as discussed in the following proposition.

**Proposition 1** *The optimization in (2) is jointly convex in $(X, z)$. Further, $\forall \gamma > 0$, $(\lambda, \gamma\epsilon)$ and $(\gamma^{-1}\lambda, \epsilon)$ lead to equivalent estimators, specifically $\widehat{\mathcal{X}}(\lambda, \gamma\epsilon) = \gamma^{-1}\widehat{\mathcal{X}}(\gamma^{-1}\lambda, \epsilon)$.*

Since, positive scaling of $\widehat{\mathcal{X}}$ preserves the resultant preference order, using Proposition 1 without loss of generality, only one of $\epsilon$ or $\lambda$ requires tuning with the other remaining fixed.

# 4 Optimization Algorithm

The optimization problem in (2) is jointly convex in $(X, z)$. Further, we later show that the proximal operator of the non–differential component of the estimate $\lambda\|X\|_* + \sum_j \mathbb{I}(z_{\Omega_j} \in \mathcal{R}_{\downarrow\epsilon}^{n_j}(y^{(j)}))$ is efficiently computable. This motivates using the proximal gradient descent algorithm [30] to jointly update $(X, z)$. For an appropriate step size $\alpha = 1/2$ and the resulting updates are as follows:

- **$X$ Update: Singular Value Thresholding** The proximal operator for $\tau\|.\|_*$ is the singular value thresholding operator $\mathcal{S}_\tau$. For $X$ with singular value decomposition $X = U\Sigma V$ and $\tau \geq 0$, $\mathcal{S}_\tau(X) = Us_\tau(\Sigma)V$, where $s_\tau$ is the soft thresholding operator given by $s_\tau(x)_i = \max\{x_i - \tau, 0\}$.
- **$z$ Update: Parallel Projections** For hard constraints on $z$, the proximal operator at $v$ is the Euclidean projection on the constraints given by $z \leftarrow \operatorname{argmin}_z \|z - v\|_2^2$, s.t. $z_{\Omega_j} \in \mathcal{R}_{\downarrow\epsilon}^{n_j}(y^{(j)}) \; \forall j \in [d_2]$. These updates decouple along each entity (column) $z_{\Omega_j}$ and can be trivially parallelized. Efficient projections onto $\mathcal{R}_{\downarrow\epsilon}^{n_j}(y^{(j)})$ are discussed Section 4.1.

---

**Algorithm 1** Proximal Gradient Descent for (2) with input $\Omega, \{y_j^{(j)}\}, \epsilon$ and paramter $\lambda$

---

**for** $k = 0, 1, 2, \ldots,$ **Until** (stopping criterion)

$$X^{(k+1)} = \mathcal{S}_{\lambda/2}\Big(X^{(k)} + \tfrac{1}{2}\big(\mathcal{P}_\Omega^*(z^{(k)} - X_\Omega^{(k)})\big)\Big), \tag{3}$$

$$\forall j, z_{\Omega_j}^{(k+1)} = \operatorname{Proj}_{\mathcal{R}_{\downarrow\epsilon}^{n_j}(y_j)}\left(\frac{z_{\Omega_j}^{(k)} + X_{\Omega_j}^{(k)}}{2}\right). \tag{4}$$

---

## 4.1 Projection onto $\mathcal{R}_{\downarrow\epsilon}^n(y)$

We begin with the following definitions that are used in characterizing $\mathcal{R}_{\downarrow\epsilon}^n(y)$.

**Definition 2 (Adjacent difference operator)** The adjacent difference operator in $\mathbb{R}^n$, denoted by $\boldsymbol{D_n} : \mathbb{R}^n \to \mathbb{R}^{n-1}$ is defined as $(\boldsymbol{D_n}x)_i = x_i - x_{i+1}$, for $i \in [n-1]$.

**Definition 3 (Incidence Matrix)** For a directed graph $\mathcal{G}(V, E)$, the incidence matrix $A_\mathcal{G} \in \mathbb{R}^{|V| \times |E|}$ is such that: if the $j^{\text{th}}$ directed edge $e_j \in E$ is from $i^{\text{th}}$ node to $k^{\text{th}}$ node, then $(A_\mathcal{G})_{ij} = 1$, $(A_\mathcal{G})_{kj} = -1$, and $(A_\mathcal{G})_{lj} = 0, \forall l \neq i$ or $k$.

Projection onto $\mathcal{R}_{\downarrow\epsilon}^n(y)$ is closely related to the *isotonic regression* problem of finding a univariate least squares fit under consistent order constraints (without margins). This isotonic regression problem in $\mathbb{R}^n$ can be solved exactly in $\mathcal{O}(n)$ complexity using the classical Pool of Adjacent Violators (PAV) algorithm [15; 4] as:

$$\text{PAV}(v) = \operatorname*{argmin}_{z' \in \mathbb{R}^n} \|z' - v\|^2 \text{ s.t. } z_i' - z_{i+1}' \leq 0. \tag{5}$$

As we discuss, simple adaptations of isotonic regression can be used for projection onto $\epsilon$-margin isotonic sets for the three special cases of interest as summarized in Table 1.

**(A) Strict Total Order:** $y_{(1)} < y_{(2)} < \cdots y_{(n)}$
In this setting, the constraint set can be characterized as $\mathcal{R}_{\downarrow\epsilon}^n(y) = \{x : \boldsymbol{D_n}x \leq -\epsilon\mathbf{1}\}$, where $\mathbf{1}$ is a vector of ones. For this case projection onto $\mathcal{R}_{\downarrow\epsilon}^n(y)$ differs from (5) only in requiring an $\epsilon$–separation and a straight forward extension of the PAV algorithm [4] can be used. Let $\boldsymbol{d}^{\text{sl}} \in \mathbb{R}^n$ be any vector such that $\mathbf{1} = -\boldsymbol{D_n}\boldsymbol{d}^{\text{sl}}$, then by simple substitutions, $\operatorname{Proj}_{\mathcal{R}_{\downarrow\epsilon}^n(y)}(x) = \text{PAV}(x - \epsilon\boldsymbol{d}^{\text{sl}}) + \epsilon\boldsymbol{d}^{\text{sl}}$.

**(B) Blockwise Total Order:** $y_{(1)} \leq y_{(2)} \leq \ldots \leq y_{(n)}$
This is a common setting for supervision in many preference completion applications, where the listwise ranking preferences obtained from ratings over discrete quantized levels $1, 2, \ldots, K$, with $K \ll n$ are prevalent. Let $y$ be partitioned into $K \leq n$ blocks $P = \{P_1, P_2, \ldots P_K\}$, such that the entries of $y$ within each partition are equal, and the blocks themselves are strictly ordered,

$$\text{i.e., } \forall k \in [K], \, \sup y(P_{k-1}) < \inf y(P_k) = \sup y(P_k) < \inf y(P_{k+1}),$$

where $P_0 = P_{K+1} = \phi$, and $y(P) = \{y_i : i \in P\}$.

Let $\boldsymbol{d}^{\text{bl}} \in \mathbb{R}^n$ be such that $\boldsymbol{d}^{\text{bl}}_{\boldsymbol{i}} = \sum_{k=1}^{K} k \,\mathbb{I}_{i \in P_k}$ is a vector of block indices $\boldsymbol{d}^{\text{bl}} = [1, 1, ..|2, 2, ..|K, K, .., K]^\top$. Let $\Pi_P$ be a set of valid permutations that permute entries only within blocks $\{P_k \in P\}$, then $\mathcal{R}^n_{\downarrow\epsilon}(y) = \{x : \exists \pi \in \Pi_P, \boldsymbol{D_n}\pi(x) \leq -\epsilon \boldsymbol{D_n} \boldsymbol{d}^{\text{bl}}\}$. We propose the following steps to compute $\widehat{z} = \text{Proj}_{\mathcal{R}^n_{\downarrow\epsilon}(y)}(x)$ in this case:

$$\text{Step 1. } \pi^*(x) \text{ s.t. } \forall k \in [K], \ \pi^*(x)_{P_k} = \text{sort}(x_{P_k})$$
$$\text{Step 2. } \widehat{z} = PAV(\pi^*(x) - \epsilon \boldsymbol{d}^{\text{bl}}) + \epsilon \boldsymbol{d}^{\text{bl}}. \tag{6}$$

The correctness of (6) is summarized by the following Lemma.

**Lemma 2** *Estimate $\widehat{z}$ from (6) is the unique minimizer for*
$$\underset{z}{argmin}\|z - x\|_2^2 \ s.t. \ \exists \pi \in \Pi_P : \boldsymbol{D_n}\pi(z) \leq -\epsilon \boldsymbol{D_n} \boldsymbol{d}^{bl}.$$

**(C) Arbitrary DAG**: $y = \mathcal{G}([n], E)$

An arbitrary DAG (not necessarily connected) can be used to represent *any* consistent order constraints over its vertices, e.g., partial rankings consolidated from multiple listwise/pairwise scores. In this case, the $\epsilon$–margin isotonic set is given by $\mathcal{R}^n_{\downarrow\epsilon}(y) = \{x : A_{\mathcal{G}}^\top x \leq -\epsilon \mathbf{1}\}$ (c.f. Definition 3). Consider $\boldsymbol{d}^{\text{DAG}} \in \mathbb{R}^n$ such that $i^{\text{th}}$ entry $\boldsymbol{d}^{\text{DAG}}_{\boldsymbol{i}}$ is the length of the longest directed chain connecting the topological descendants of the node $i$. It can be easily verified that, the isotonic regression algorithm for arbitrary DAGs applied on $x - \epsilon \boldsymbol{d}^{\text{DAG}}$ gives the projection onto $\mathcal{R}^n_{\downarrow\epsilon}(y)$. In this most general setting, the best isotonic regression algorithm for exact solution requires $\mathcal{O}(nm^2 + n^3 \log n^2)$ computation [34], where $m$ is the number of edges in $\mathcal{G}$. While even in the best case of $m = o(n)$, the computation can be prohibitive, we include this case for completeness. We also note that this case of partial DAG ordering cannot be handled in the standard matrix completion setting without consolidating the partial ranks to total order.

| | $\mathcal{R}^n_{\downarrow\epsilon}(y)$ | $\text{Proj}_{\mathcal{R}^n_{\downarrow\epsilon}(y)}(x)$ | Computation |
|---|---|---|---|
| (A) | $\{x : \boldsymbol{D_n} x \leq -\epsilon \mathbf{1}\}$ | $PAV(x - \epsilon \boldsymbol{d}^{\text{sl}}) + \epsilon \boldsymbol{d}^{\text{sl}}$ | $\mathcal{O}(n)$ |
| (B) | $\{x : \exists \pi \in \Pi_P, \boldsymbol{D_n}\pi(x) \leq -\epsilon \mathbf{1}\}$ | $\pi_P^{*^{-1}}\left(PAV(\pi_P^*(x) - \epsilon \boldsymbol{d}^{\text{bl}}) + \epsilon \boldsymbol{d}^{\text{bl}}\right)$ | $\mathcal{O}(n \log n)$ |
| (C) | $\{x : A_{\mathcal{G}}^\top x \leq -\epsilon \mathbf{1}\}$ | $\text{IsoReg}(x - \epsilon \boldsymbol{d}^{\text{DAG}}, \mathcal{G}) + \epsilon \boldsymbol{d}^{\text{DAG}}$ [34] | $\mathcal{O}(n^2 m + n^3 \log n)$ |

Table 1: Summary of algorithms for $\text{Proj}_{\mathcal{R}^n_{\downarrow\epsilon}(y)}(x)$

## 4.2 Computational Complexity

It can be easily verified that gradient of $\frac{1}{2}\|\mathcal{P}_\Omega(X) - z\|_2^2$ is 2–Lipschitz continuous. Thus, from standard results on convergence proximal gradient descent [30], Algorithm 1, converges to within an $\epsilon$ error in objective in $\mathcal{O}(1/\epsilon)$ iterations. Compared to proximal algorithms for standard matrix completion [5; 27], the additional complexity in Algorithm 1 arises in the $z$ update (4), which is a simple substitution $z^{(k)} = X^{(k)}_\Omega$ in standard matrix completion. For total orders, the $z$ update of (4) is highly efficient and is asymptotically within an additional $\log |\Omega|$ factor of the computational costs for standard matrix completion.

## 5 Generalization Error

Recall that $y_j$ are (noisy) partial rankings of subset of items for each user, obtained from $g_j(\Theta_j^* + W_j)$ where $W$ is a noise matrix and $g_j$ are unknown and arbitrary transformations that only preserve that ranking order within each column. The estimator and the algorithms described so far are independent of the sampling distribution generating $(\Omega, \{y_j\})$. In this section we quantify simple generalization error bounds for (2).

**Assumption 1 (Sampling $(\mathbb{P}_\Omega)$)** For a fixed $W$ and $\Theta^*$, we assume the following sampling distribution. Let be $c_0$ a fixed constant and $R$ be pre–specified parameter denoting the length of single listwise observation. For $s = 1, 2, \ldots, |S| = c_0 d_2 \log d_2$,

$$j(s) \sim \text{uniform}[d_2], \quad \mathcal{I}(s) \sim \text{randsample}([d_1], R),$$
$$\Omega(s) = \{(i, j(s)) : i \in \mathcal{I}(s)\}, \quad y(s) = g_{j(s)}(\mathcal{P}_{\Omega(s)}(\Theta^* + W)). \tag{7}$$

Further, we define the notation: $\forall j, \mathcal{I}_j = \bigcup_{s:j(s)=j} \mathcal{I}(s), \quad \Omega_j = \bigcup_{s:j(s)=j} \Omega(s),$ and $n_j = |\Omega_j|.$

For each column $j$, the listwise scores $\{y(s) : j(s) = j\}$ jointly define a consistent partial ranking of $\mathcal{I}_j$ as the scores are subsets of a monotonically transformed preference vector $g_j(\Theta_j^* + W_j)$. This consistent ordering is represented by a DAG $y^{(j)} = \text{PartialOrder}(\{y(s) : j(s) = j\})$. We also note that $\mathcal{O}(d_2 \log d_2)$ samples ensures that each column is included in the sampling with high probability.

**Definition 4 (Projection Loss)** *Let* $y = \mathcal{G}([n], E)$ *or* $y \in \mathbb{R}^n$ *define a partial ordering or total order in* $\mathbb{R}^n$, *respectively. We define the following convex surrogate loss over partial rankings.*

$$\Phi(x, y) = \min_{z \in \mathcal{R}_{\downarrow\epsilon}^n(y)} \|x - z\|_2$$

**Theorem 3 (Generalization Bound)** *Let* $\widehat{X}$ *be an estimate from* (2). *With appropriate scaling let* $\|\widehat{X}\|_F = 1$ , *then for constants* $K_1$ $K_2$, *the following holds with probability greater than* $1 - \delta$ *over all observed rankings* $\{y^{(j)}, \Omega_j : j \in [d_2]\}$ *drawn from* (7) *with* $|S| \geq c_0 d_2 \log d_2$:

$$\mathbb{E}_{y(s),\Omega(s)} \Phi(\widehat{X}_{\Omega(s)}, y(s)) \leq \frac{1}{|S|} \sum_{s=1}^{|S|} \Phi(\widehat{X}_{\Omega(s)}, y(s)) + K_1 \frac{\|\widehat{X}\|_* \log^{1/4} d}{\sqrt{d_1 d_2}} \sqrt{\frac{d \log d}{R|S|}} + K_2 \sqrt{\frac{\log 2/\delta}{|S|}}.$$

Theorem 3 quantifies the test projection loss over a random $R$ length items $\mathcal{I}(s)$ drawn for a random entity/user $j(s)$. The bound provides a trade–off between observable training error and complexity defined by nuclear norm of the estimate.

# 6 Experiments

We evaluate our model on two collaborative preference estimation tasks: $(a)$ a standard user-item recommendaion task on a benchmarked dataset from Movielens, and $(b)$ identifying associations between brain–regions and cognitive terms using the neurosynth dataset [37].

**Baselines:** The following baseline models are compared in our experiments:

- *Retargeted Matrix Completion (RMC)*: the estimator proposed in (2).
- *Standard Matrix Completion (SMC) [8]*: We primarily compare our estimator with the standard convex estimator for matrix completion using nuclear norm minimization.
- *Collaborative Filtering Ranking CoFi-Rank [35]*: This work addresses collaborative filtering task in a listwise ranking setting.

For SMC and MRPC, the hyperparameters were tuned using grid search on a logarithmic scale. Due to high computational cost with tuning parameters in CofiRank, we use the code and default parameters provided by the authors.

**Evaluation metrics:** The performance on preference estimation tasks are evaluated on four ranking metrics: $(a)$ Normalized Discounted Cummulative Gains (NDCG@N), $(b)$ Precision@N, $(c)$ Spearmann Rho, and $(d)$ Kendall Tau, where the later two metrics measure the correlation of the complete ordering of the list, while the former two metrics primarily evaluate the correctness of ranking in the top of the list (see Liu et. al. [25] for further details on these metrics).

**Movielens dataset (blockwise total order)** Movielens is a movie recommendation website administered by GroupLens Research. We used competitive benchmarked movielens 100K dataset. We used the 5–fold train/test splits provided with the dataset (the test splits are non-overlapping). We discarded a small number of users that had less than 10 ratings in any of 5 training data splits. The resultant dataset consists of 923 users and 1682 items. The ratings are blockwise ordered – taking one of 5 values in the set $\{1, 2, \ldots, 5\}$. During testing, for each user, the competing models return a ranking of the test-items, and the performance is averaged across test-users. Table 2 presents the results of our evaluation averaged across 5 train/test splits on the Movielens dataset, along with the standard deviation. We see that the proposed retargeted matrix completion (RMC) significantly and consistently outperforms SMC and CoFi-Rank [35] across ranking metrics.

|           | NDCG@5          | Precision@5      | Spearman Rho     | Kendall Tau      |
|-----------|-----------------|------------------|------------------|------------------|
| **RMC**   | **0.7984(0.0213)** | **0.7546(0.0320)** | **0.4137(0.0099)** | **0.3383(0.0117)** |
| SMC       | 0.7863(0.0243)  | 0.7429(0.0295)   | 0.3722(0.0106)   | 0.3031(0.0117)   |
| CoFi-Rank | 0.7731(0.0213)  | 0.7314(0.0293)   | 0.3681(0.0082)   | 0.2993(0.0110)   |

Table 2: Ranking performance for recommendations in Movielens 100K. Table shows mean and standard deviation over 5 fold train/test splits. For all reported metrics, higher values are better [25].

**Neurosynth Dataset (almost total order)**   Neurosynth[37] is a publicly available database consisting of data automatically extracted from a large collection of functional magnetic resonance imaging (fMRI) publications (11,362 publications in current version). For each publication , the database contains the abstract text and all reported 3-dimensional peak activation coordinates in the study. The text is pre-processed to remove common stop-words, and any text with less than .1% frequency, leaving a total of 3169 terms. We applied the standard brain map to the activations, removing voxels outside of the grey matter. Next the activations were downsampled from $2mm^3$ voxels to $10mm^3$ voxels using the nilearn python package, resulting in a total of 1231 dense voxels. The affinity measure between 3169 terms and 1231 consolidated voxels is obtained by multiplying the term $\times$ publication and the publication $\times$ voxels matrices. The resulting data is dense high-rank preference matrix. With very few tied preference values, this setting best fits the case of total ordered observations (case A in Section 4.1). Using this data, we consider the reverse inference task of ranking cognitive concepts (terms) for each brain region (voxel) [37].

*Train-val-test:* We used 10% of randomly sampled entries of the matrix as test data and a another 10% for validation. We created training datasets at various sample sizes by subsampling from the remaining 80% of the data. This random split is replicated multiple times to obtain 3 bootstrapped datasplits (note that unlike cross validation, the test datasets here can have some overlapping entries).

The results in Fig. 1 show that the proposed estimate from (2) outperforms standard matrix completion in terms of popular ranking metrics.

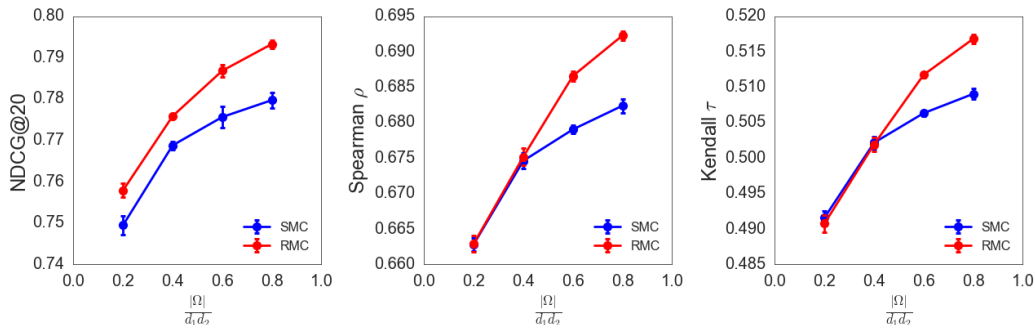

Figure 1: Ranking performance for reverse inference in Neurosynth data. x-axis denotes the fraction of the affinity matrix entries used as observations in training. Plots show mean with errorbars for standard deviation over 3 bootstrapped train/test splits. For all the reported ranking metrics, higher values are better[25].

## 7   Conclusion

Our work addresses the problem of collaboratively ranking; a task of growing importance to modern problems in recommender systems, large scale meta-analysis, and related areas. We proposed a novel convex estimator for collaborative LETOR from sparsely observed preferences, where the observations could be either score vectors representing total order, or more generally directed acyclic graphs representing partial orders. Remarkably, in the case of complete order, the complexity of our algorithm is within a $\log$ factor of the state–of–the–art algorithms for standard matrix completion. Our estimator was empirically evaluated on real data experiments.

**Acknowledgments**   SG and JG acknowledge funding from NSF grants IIS-1421729 and SCH 1418511.

## Footnotes

[1]http://neurosynth.org/

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
