[Supplementary Material]


# A Estimator and Algorithm

## A.1 Proof of Proposition 1

**Statement of the Proposition:** The optimization in (2) is jointly convex in $(X, z)$. Further, $\forall \gamma > 0$, $(\lambda, \gamma\epsilon)$ and $(\gamma^{-1}\lambda, \epsilon)$ lead to equivalent estimators, specifically $\widehat{\mathcal{X}}(\lambda, \gamma\epsilon) = \gamma^{-1}\widehat{\mathcal{X}}(\gamma^{-1}\lambda, \epsilon)$.

**Proof:** Let $f_{\lambda,\epsilon}(X) = \min\limits_{z \in \mathbb{R}^{|\Omega|}} \quad \lambda\|X\|_* + \frac{1}{2}\|z - \mathcal{P}_\Omega(X)\|_2^2$ .
$$\text{s.t.} \quad \forall j, \; z_{\Omega_j} \in \mathcal{R}_{\downarrow\epsilon}^{n_j}(y^{(j)}),$$

We have,

$$
\begin{aligned}
f_{\lambda,\gamma\epsilon}(X) &= \min_z \lambda\|X\|_* + \frac{1}{2}\|z - \mathcal{P}_\Omega(X)\|_2^2 \;\; \text{s.t.} \;\; z_{\Omega_j} \in \mathcal{R}_{\downarrow\gamma\epsilon}^{n_j}(y^{(j)}), \\
&\overset{(a)}{=} \min_{\bar{z}} \lambda\|X\|_* + \frac{1}{2}\|\gamma\bar{z} - \mathcal{P}_\Omega(X)\|_2^2 \;\; \text{s.t.} \;\; \bar{z}_{\Omega_j} \in \mathcal{R}_{\downarrow\epsilon}^{n_j}(y^{(j)}), \\
&= \gamma^2\min_{\bar{z}} \frac{\lambda}{\gamma}\|X/\gamma\|_* + \frac{1}{2}\|\bar{z} - \mathcal{P}_\Omega(X/\gamma)\|_2^2 \;\; \text{s.t.} \;\; \bar{z}_{\Omega_j} \in \mathcal{R}_{\downarrow\epsilon}^{n_j}(y^{(j)}), \\
&= \gamma^2 f_{\gamma^{-1}\lambda,\epsilon}(X/\gamma),
\end{aligned}
\tag{9}
$$

where $(a)$ follows from reparameterizing the optimization using $\bar{z} = z/\gamma$ as the geometry of $\mathcal{R}_{\downarrow\gamma\epsilon}^{n_j}(y^{(j)})$ which is set of linear constraints of the form $z_i - z_k \leq \gamma\epsilon$. From above set of equations, if $X \in \text{Argmin}\limits_X f_{\lambda,\gamma\epsilon}(X)$, then $\gamma^{-1}X \in \text{Argmin}\limits_X f_{\gamma^{-1}\lambda,\epsilon}(X)$.

## A.2 Proof of Lemma 2

**Statement of the Lemma:** Consider the following steps,

$$
\begin{aligned}
&\text{Step 1. } \pi^*(x) \text{ s.t. } \forall k \in [K], \; \pi^*(x)_{P_k} = \text{sort}(x_{P_k}) \\
&\text{Step 2. } \widehat{z} = PAV(\pi^*(x) - \epsilon\boldsymbol{d}^{\text{bl}}) + \epsilon\boldsymbol{d}^{\text{bl}}.
\end{aligned}
\tag{10}
$$

Estimate $\widehat{z}$ is the unique minimizer for

$$\underset{z}{\text{argmin}}\|z - x\|_2^2 \text{ s.t. } \exists\pi \in \Pi_P : \boldsymbol{D_n}\pi(z) \leq \epsilon\boldsymbol{D_n}\boldsymbol{d}^{\text{bl}}.$$

**Proof:** A version of the lemma for linear orders was proved in [2]. In general,

$$
\begin{aligned}
&\min_z \|z - x\|_2^2 \text{ s.t. } \exists\pi \in \Pi_P : \boldsymbol{D_n}\pi(z) \leq \epsilon\boldsymbol{D_n}\boldsymbol{d}^{\text{bl}} \\
&= \min_{z,\pi \in \Pi_P} \|z - x\|_2^2 \text{ s.t. } \boldsymbol{D_n}\pi(z) \leq \epsilon\boldsymbol{D_n}\boldsymbol{d}^{\text{bl}} \\
&\overset{(a)}{=} \min_w \min_{\pi \in \Pi_P} \|\pi^{-1}(w + \epsilon\boldsymbol{d}^{\text{bl}}) - x\|_2^2 \text{ s.t. } \boldsymbol{D_n}w \\
&\leq 0 \overset{(b)}{=} \min_{w:\boldsymbol{D_n}w \leq 0} \min_{\pi \in \Pi_P} \|w + \epsilon\boldsymbol{d}^{\text{bl}} - \pi(x)\|_2^2 \\
&\overset{(c)}{=} \min_{w:\boldsymbol{D_n}w \leq 0} \|w + \epsilon\boldsymbol{d}^{\text{bl}} - \pi^*(x)\|_2^2,
\end{aligned}
\tag{11}
$$

where $\pi^*(x)$ is the update from Step 1 stated above, $(a)$ follows reparametrizing $w := \pi(z) - \epsilon\boldsymbol{d}^{\text{bl}}$, $(b)$ follows as for all permutations $\pi$ using $\|x\|_2^2 = \|\pi(x)\|_2^2$, and $(c)$ follows form Proposition 5 as $\boldsymbol{D_n}w \leq 0$ from constraints and $\epsilon\boldsymbol{D_n}\boldsymbol{d}^{\text{bl}} \leq 0$ by construction. The final minimization is solved using Step 2. $\square$

**Proposition 5** *For any sorted $z \in \mathbb{R}^n$ such $\boldsymbol{D_n}z \leq 0$, $\pi^* = \underset{\pi \in \Pi_P}{argmin}\|z - \pi(x)\|_2^2$, where $\pi^*$ is the permutation from Step* 1.

$\Pi_P$ allows for all possible permutations within each partition $P_k$. Proposition follows from optimality of sorting within each block. $\square$

 # B   Generalization Error

 ## B.1   Background

**Definition 5 (Rademacher Complexity)** *Let $X_1, X_2, \ldots, X_n \in \mathcal{X}$ be drawn iid from a distribution $\mathbb{P}_X$. For a function class $\mathcal{F} : \mathcal{X} \to \mathcal{A}$, the empirical Rademacher complexity is defined as,*

$$\widehat{\mathfrak{R}}_n(\mathcal{F}) = \mathbb{E}_\sigma \sup_{f \in \mathcal{F}} \Big( \frac{1}{n} \sum_{i=1}^n \sigma_i f(X_i) \Big),$$

 *where $\sigma_1, \sigma_2, \ldots, \sigma_n$ are iid Rademacher variables, i.e., $\pm 1$ with probability $1/2$.*

 *The Rademacher complexity with respect tp $\mathbb{P}_X$ is then defined as $\mathfrak{R}_n(\mathcal{F}) = \mathbb{E}_{\mathbb{P}_X} \widehat{\mathfrak{R}}_n(\mathcal{F})$.*

**Theorem 6 (Generalization Error Bound (Corollary 15 in [3]))** *Consider a loss function $\ell : \mathcal{Y} \times \mathbb{R}^m \to [0,1]$ and a bounded function class $\mathcal{F} : \mathcal{X} \to \mathbb{R}^m$ such that $\mathcal{F}$ is a direct sum of $\mathcal{F}_1, \mathcal{F}_2, \ldots, \mathcal{F}_m$. Further, if $\ell$ is $L$–Lipschitz continuous with respect to Euclidean distance on $\mathbb{R}^m$ and is uniformly bounded. Let $\{(X_i, Y_i), i = 1, 2, \ldots, n\}$ be sampled form a distribution $\mathbb{P}_{X,Y}$. Then there exists a constant $c$ such that, for any integer $n$ and any $\delta \in (0,1)$, with probability atleast $1 - \delta$, over all sample of length $n$, the following holds for every $f \in \mathcal{F}$:*

$$\mathbb{E}_{X,Y} \ell(Y, f(X)) \leq \frac{1}{n} \sum_{i=1}^n \ell(Y_i, f(X_i)) + cL \sum_{i=1}^m \widehat{\mathfrak{R}}_n(\mathcal{F}_m) + \sqrt{\frac{8 \log(2/\delta)}{n}}$$

 ## B.2   Proof of Theorem 4

 **Lemma 7** *$\phi(., y)$ is convex and $2$–Lipschitz continuous with respect to $\ell_2$ norm.*

 **Proof:** Convexity follows form $\Phi$ being a marginal of a convex function. For a any convex set $C$
 and its projection operator $P_C$, we have the following for all $x, x'$:

$$\big| \|x - P_C(x)\|_2 - \|x' - P_C(x')\|_2 \big| \leq \|x - P_C(x) - x' + P_C(x')\|_2$$
$$\leq \|x - x'\|_2 + \|P_C(x) - P_C(x')\|_2 \leq 2\|x - x'\|_2$$

 Consider a vector class of functions in $\mathbb{R}^R$, $\mathcal{F}_R = \{\Omega(s) \to X_{\Omega(s)} \in \mathbb{R}^R : \|X\|_* \leq M\}$, where
 $\Omega(s)$ are sampled as in the main paper. Also, consider another function classes $\mathcal{F}_{ij} = \{(i,j) \to$
 $X_{ij} : \|X\|_* \leq M\}$. It can be seen that $\mathcal{F}_R$ is an $R$ way direct sum of $\mathcal{F}_{ij}$. In order to use Theorem 6,
 we need to estimate the Rademacher complexity of $\mathcal{F}_{ij}$.

 **Lemma 8** *Let $\Omega = \cup_j \Omega_j$ obtained from combining samples form Assumption 1. The distribution of*
 *$\Omega$ is equivalent to uniformly sampling with replacement $|\Omega| = c_0 d_2 R \log d_2$ entries from $[d_1] \times [d_2]$.*

 *Proof*: For $k = 1, 2 \ldots |\Omega|$, $\forall (i,j) \in [d_1] \times [d_2]$,
 $\mathbb{P}((i,j) = \Omega_k) = \frac{1}{d_1 d_2}$.

 Thus, given $(i,j) \in [d_1] \times [d_2]$, $\mathbb{P}((i,j) \in \Omega) = \frac{|\Omega|}{d_1 d_2}$. $\qquad\qquad\qquad\qquad\qquad\qquad\qquad \square$

 **Lemma 9 (Theorem 29 in [31])** *For a universal constant $K$, the Rademacher complexity of matri-*
 *ces in $\mathbb{R}^{d_1 \times d_2}$ of trace norm $M$, over uniform sampling of index pairs $\Omega$ is bounded by the following*
 *whenever $|\Omega| > d \log d$*

$$\mathfrak{R}(\{\|X\|_* \leq M\}) \leq K \frac{M \log^{1/4} d}{\sqrt{d_1 d_2}} \sqrt{\frac{d \log d}{|\Omega|}} \tag{12}$$

 From Lemma 8, it can be seen that Lemma 9 applies to samples drawn according to Assumption 1.

 For the function class $\mathcal{F}_R = \{\Omega(s) \to X_{\Omega(s)} : \|X\|_* \leq M\}$, for some $M$. The theorem now
 follows by using the Rademacher complexity bound in Lemma 9 and Lipschitz continuity of $\Phi(., y)$
 from 7 in Theorem 6.