[Reviews · NeurIPS 2016]

Reviewer 1

Summary

The submission proposes a new approach for preference completion. Frequently used methods are based on low rank matrix completion or the `learning to rank' formulation, both of which rely heavily on the scores provided by the user. In contrast, the proposed approach does not directly rely on scores but rather on rankings between pairs of elements. Hence scores can be transformed in an arbitrary but monotonic way. More generally, the proposed approach is also directly applicable to given partial rankings, without any transformation to scores. To this end, a set of vectors is defined which fulfill the provided ordering with at least an epsilon-margin. The estimator for the preference matrix is then obtained by minimizing its nuclear norm and its deviation from the entries in the set of vectors fulfilling the provided ordering. A proximal gradient descent algorithm is suggested which results in two steps, (i) a singular value thresholding when optimizing w.r.t. the preference matrix, and (ii) a projection onto the feasible set of valid rankings. The latter projection is equivalent to isotonic regression, hence it can be solved using the `Pool Adjacent Violators' algorithm if we consider strict or blockwise total ordering of preferences. A more complex algorithm has to be used in case we are given an arbitrary (consistent) ordering. The algorithmic complexity of the proposed algorithm is composed out of two parts, (i) singular value thresholding, which is also required in standard matrix completion techniques, and (ii) projection onto the feasible set, which is feasible in simple ordering cases but expensive generally. The proposed approach is evaluated on the Movielens dataset and a neurosynth dataset and shown to outperform simple baselines such as standard low-rank matrix completion and Cofi-Rank.

Qualitative Assessment

Review summary: -------- The proposed technique does not directly take into account the provided preference scores which is a clear advantage since scores may often be noisy. However the proposed approach is generally very expensive (an experimental comparison hasn't been shown) and it might not handle inconsistent ordering very well. The set of feasible vectors may indeed be empty and no mechanism is provided for the algorithm to fail gracefully. Technical quality: the feasible set might be empty if confusing rankings are provided as input, which hasn't been analyzed Novelty: an interesting technique for ranking Impact: a comparison with recent methods is missing Clarity: well described generally Review details: -------- 1. The defined set of vectors fulfilling the provided ordering may be empty or contain only a few elements if the provided groundtruth ordering is not consistent. The proposed approach does not seem to fail gracefully in this setting, i.e., a confusing ranking may actually remove good solutions. Did the authors investigate the properties of the proposed algorithm in this setting? 2. While Lemma 2 is interesting, it's important to relate it back to the original problem. As a reader I'm therefore wondering whether the proposed approach still optimizes the original formulation. 3. Some more intuition could according to my opinion be provided by shortining the intro. In particular the description of the generalization bound is very short and follows from existing literature. This could be stated more clearly. Was the symbol `d' defined? 4. In the experimental evaluation, classical baselines are employed. As a reader I'm wondering about more recent approaches and a comparison w.r.t. the proposed algorithm. Can the authors comment? 5. Ideally it should also be evaluated whether it's worth to spend the additional effort of projecting onto the feasible set. An evaluation of the computation time would be desirable. Minor comments: -------------- - typo in Eq. (2)? Shouldn't there be a minimization over z? - possible typo in Prop. 1? It seems like the second \gamma in the last equation shouldn't be inverted. Didn't check completely though. - possible typo in Lemma 2: a minus sign may be missing in the characterization of the constraint set to make it consistent with the definition of the feasible set in line 218. - the writing should be corrected here and there

Confidence in this Review

2-Confident (read it all; understood it all reasonably well)


Reviewer 2

Summary

This paper addresses the problem of fitting a low-rank matrix that preserves the ordering within each row (or each column) from the partially observed entries. A similar problem has been studied in SIMs with low-rank structure, where the main difference is in what is to be estimated. In this paper, we are only interested in retrieving the low-rank matrix that preserves the orderings, and not necessarily the monotonic transformation that resulted in the observed entries. A standard mean-squared error with nuclear-norm regularization is minimized, with the constraints specifying the orderings to be preserved (up to an epsilon margin). Generalization error bound for the solution to this optimization is provided, along with numerical experiments suggesting the superiority of the proposed approach.

Qualitative Assessment

The problem formulation is not entirely new, since it has been studied in [12]. the main novelty in the formulation is that this paper focuses on recovering the underlying low-rank preference matrix, only. Once the problem has been formulated this way, the authors propose minimizing the nuclear norm to encourage low-rank solutions, over the \epsilon-margin isotonic set to preserve the ordering. The formulation in (2) is nice in the sense that it is a convex minimization. A downside is that there is no discussion on how to choose \lambda. The theoretical guarantees in the generalization error gives no insight into how to choose \lambda, and this should be explained as such. Having a hyper parameter in the algorithm is acceptable, but it should be clearly stated that currently there is no systematic approach to choose this, other than perhaps cross validation (if you have some training data) or application specific side information. In the generalization error bound in Theorem 4, it is not clear what role the rank of $\Theta^*$ plays. It seems to hold generically for all $\Theta^*$, low-rank or not. Plugging-in a trivial bound $\|\hat{X}\}_* \leq \|\hat{X}\|_F \sqrt{d}$, it seems like one can get arbitrarily small generalization error bound with effective sample size $R|S| = O(d \log d)$. Now, consider a scenario where $\Theta^*$ is full-rank. Although there are d^2 parameters to be learned, because the metric $\Phi()$ effectively reduces the size of the space to $R|S|$, we only need $R|S| = O (d log d)$. Recall that since there is no low-rank structure, each column could have been estimated separately, without the nuclear norm minimization, to get the same error bound. This is just an observation on the statement of Theorem 4, and we would expect it to give us something tighter for low-rank $\Theta^*$. However, it seems like the upper bound on RHS of Theorem 4 is not necessarily tighter, since we do not have any reasonable bound on the nuclear norm of $\hat{\Theta}$. It is well known that often times we get full-rank solutions for solving nuclear norm minimization. When the resulting estimate has a low nuclear norm in comparisons to $\|\hat{\Theta}\|_F \sqrt{d}$, then the theorem is tighter than full-rank case as expected. This point should be made clear, and resulting nuclear norm of the estimates should be evaluated numerically for some synthetic scenarios. Minor comments: - On page 4, in Eq. (2), $\min_x$ should be $min_z$. - A missing reference: "Individualized rank aggregation using nuclear norm regularization", by Y Lu, SN Negahban

Confidence in this Review

3-Expert (read the paper in detail, know the area, quite certain of my opinion)


Reviewer 3

Summary

The paper is in the setting of collaborative filtering, where the main objective is not to get the exact scores of the users right, but rather to get the rankings of the items for each user right. The authors propose a new convex optimization based algorithm for this that is very natural in a sense. The algorithm essentially considers an intermediate variable that has the same ordering as the given scores, and optimizes a model that fits to this intermediate variable along with the intermediate variable itself -- the optimization objective is jointly convex in both the variables. Essentially the usage of this intermediate variable overcomes the difficulty of different people having different baselines and levels of leniency. The standard low-rank assumption on the score matrix is replaced by a low rank assumption on some monotone transformation of the column of the score matrix. The algorithm itself proceeds in an iterative fashion optimizing the intermediate variable and the model alternatively. The intermediate variable optimization step has nice links to isotonic regression, and allows the usage of algorithms for that purpose.

Qualitative Assessment

The paper is simply and clearly written. The approach proposed by the authors is very natural and simple, and in hindsight looks to be the obvious approach to follow for the setting in the paper. This paper would have very good reference value for future researchers before they go on to propose any more complicated algorithms. Minor comments: It would be good to mention that even vanilla matrix factorization methods can incorporate differing user baselines and leniency, but only linear transformations. The computational complexity section could be expanded in a little more detail giving the explicit computational comlpexity for achieving a solution that is $\epsilon$-close to the solution of the optimization problem. The generalization bound in Theorem 4, is a little bit unsatisfying, it would have been better if the projection error of the estimate with $\Theta^*$ were bounded directly. Also a lower bound of sorts showing this is the best result possible would have been nice. In Equation 2, (line 173) , the optimization variable should be $z$.

Confidence in this Review

2-Confident (read it all; understood it all reasonably well)


Reviewer 4

Summary

This paper considers the problem of “collaborative preference completion” from partial rankings, which could be in the form of either ordinal scores/ratings or pairwise comparisons among some subset of items. Specifically, consider a set of users and a set of items, and say that each user expresses preferences among some subset of the items by either assigning ordinal scores/ratings to them or by providing pairwise comparison judgments for some of the pairs. From this, it is desired to estimate a ranking of the remaining items for each user. In the case of user preferences expressed as ordinal scores/ratings, a common approach is to use a low rank matrix completion algorithm to estimate the missing scores, and to rank the remaining items for each user based on these scores. Usually, the criterion in estimating the matrix is to minimize the squared error over the observed entries. The paper instead advocates an approach where the criterion is more closely tied to the ranking performance measures eventually used in evaluation. Specifically, the paper proposes finding a low rank matrix such that the resulting pairwise comparison outcomes on the observed item pairs are mostly similar to the observed comparisons. This is formulated as an optimization problem that is jointly convex in two sets of parameters, and a proximal gradient descent based algorithm is proposed to solve it. A generalization error bound is provided under a simple sampling model. The paper contains results of experiments on two publicly available real data sets: (1) the MovieLens 100K collaborative filtering data set, which contains an incomplete user-movie ratings matrix, and on which the proposed approach is shown to outperform standard matrix completion and another baseline COFI-Rank, and (2) the neurosynth data set for a different type of problem in which the input is an incomplete matrix indexed by brain regions in one dimension and cognitive terms in the other dimension, with entries containing the number of times a term has been associated with a brain region in neuroscience publications; here the goal is to rank, for each brain region, the cognitive terms according to their frequency of association with the region, and the proposed approach is again shown to outperform standard matrix completion.

Qualitative Assessment

The approach of the paper makes sense given that evaluation in many of these problems focuses not on the estimated matrix entries themselves but rather on the quality of the resulting rankings. The experiments also appear to corroborate this (but see questions below). The paper is reasonably well written overall, although it is a little dense to read in some places. Questions about experiments that should be answered in the author rebuttal: 1. How do the runtimes of the proposed algorithm compare with standard matrix completion on the two data sets (and COFI-Rank on the first)? 2. neurosynth data: what are the numbers in this matrix like (range, mean, std) etc (and how many publications is it based on)? What is its sparsity level? How was the data divided into train/test splits (i.e. what are y-axis measures in Fig 1 computed on?) ---- After author response ---- I have read the author response and it addresses most of my concerns (if the paper is accepted, the authors should include more details of the neurosynth data and experiments as promised).

Confidence in this Review

3-Expert (read the paper in detail, know the area, quite certain of my opinion)


Reviewer 5

Summary

The paper proposes a method for completing a partial set of (entity, item) preference rankings under a low rank assumption on the full preference matrix. This work builds on previous ranking-loss-based methods in that the known preferences are assumed only to convey a relative order. The proposed method, based on the technique of monotone retargeting, iteratively optimizes the preference matrix, leaving the known preference scores fixed, and monotonically transforming the preference scores in order to improve the fit. Experiments on the Movielens dataset show the proposed method to moderately outperform a standard method, and on the Neurosynth dataset to significantly outperform a standard method. The authors also prove a generalization error bound.

Qualitative Assessment

It is unclear whether the baseline methods against which the proposed method is compared are state of the art. Typo on line 71: neurosyth should be neurosynth

Confidence in this Review

1-Less confident (might not have understood significant parts)


Reviewer 6

Summary

This work studied the preference completion problem, which mainly suggested to achieve personalized recommendation by recovering the relative preferences instead of the exact affinity scores for the items. It applied the idea of a listwise learning to rank algorithm, i.e. monotone retargeting, to the recommendation problem. A monotone retargeted low rank estimator was proposed. And efficient algorithms for solving the optimization problem were presented. During evaluation, it also applied the algorithm to a neuroimaging meta-analysis problem.

Qualitative Assessment

This work can be seen as an extension of the learning to rank idea to the collaborative filtering problem, which was not brand new. Some other works should have already exploited this idea although different learning to rank algorithms may have been used. More comparison between the proposed method and similar algorithms should be carried out to validate its performance. In Section 4.1, three cases of the observed partial ranking were discussed and different projection algorithms were proposed correspondingly. However, not all of these algorithms were evaluated through experiments. There were only two datasets used in experiments and the authors didn’t state which cases each dataset belonged to. For the two baselines, it is not clear which specific algorithm the standard matrix completion (SMC) stands for. Some references or more descriptions about it should be added. The result of the COFI-Rank algorithm, which was proposed in 2008, was not able to represent the state of the art. More recent works especially those that also exploited the idea of learning to rank for the recommending problem should be compared, e.g. Park et al.’s work in 2015 [29].

Confidence in this Review

1-Less confident (might not have understood significant parts)